# Framework of an IoT-based Industrial Data Management for Smart Manufacturing

**Muhammad Saqlain [1], Minghao Piao [2], Youngbok Shim [3] and Jong Yun Lee [1], ***

[1]   Department of Computer Science, Chungbuk National University, Cheongju, Chungbuk 28644, Korea;
     m.saqlain1240@yahoo.com

[2]   Department of Smart Factory Management, Chungbuk National University, Cheongju, Chungbuk 28644,
     Korea; myunghopark@gmail.com

[3]   Department of Digital Information Convergence, Chungbuk National University, Cheongju, Chungbuk
     28644, Korea; ybshim@weruntech.com

*   Correspondence: jongyun@chungbuk.ac.kr; Tel.: +82(0)-43-261-2789

**Abstract:** The Internet of Things (IoT) is the global network of interrelated physical devices such as sensors, actuators, smart applications, objects, computing devices, mechanical machines, and people that are becoming an essential part of the internet. In an industrial environment, these devices are the source of data which provide abundant information in manufacturing processes. Nevertheless, the massive, heterogeneous, and time-sensitive nature of the data brings substantial challenges to the real-time collection, processing, and decision making. Therefore, this paper presents a framework of an IoT-based Industrial Data Management System (IDMS) which can manage the huge industrial data, support online monitoring, and control smart manufacturing. The framework contains five basic layers such as physical, network, middleware, database, and application layers to provide a service-oriented architecture for the end users. Experimental results from a smart factory case study demonstrate that the framework can manage the regular data and urgent events generated from various factory devices in the distributed industrial environment through state-of-the-art communication protocols. The collected data is converted into useful information which improves productivity and the prognosis of production lines.

**Keywords:** industrial internet of things (IIoT); smart factory; data management framework; IoT middleware; factory floor

## 1. Introduction

The Industrial Internet of Things (IIoT), smart factory, and smart manufacturing are the applications of the IoT that focus on cheap, smart, and small size interconnected factory devices [1]. These can be defined as the collection of machines contexts, objects, people, artificial intelligence, advanced analytics, and cloud computing to increase productivity by improving the health of machines [2]. The IIoT is enabling for transmission of real-time data across the industrial network to create manufacturing intelligence [3]. The IIoT is not just a single technology but a combination of different techniques such as Internet of Things (IoT), big data, cyber physical systems (CPS), machine learning (ML), and simulation, to organize smart operations in the industrial environments [4]. This combination provides promising solutions for real-time monitoring activities. The IIoT produces massive industrial data which can be used for prognosis, predict, and control the manufacturing system. The information obtained from the raw industrial data enables the manufacturer to recognize the unconventional applications, provides a valuable source of strategic opportunities, and helps for organizing smarter business decisions. Consequently, the IIoT can detect special events or a fault occurring in a real-time manufacturing process without human-centric and manual operations. In [5], the authors proposed a knowledge-based sensors failure identification and prevention model. Thus,

early fault detection and failure identification eliminate redundant production breaks, increase profit by an integrated monitoring system, predict machines health status, and reduce maintenance cost and decision-making complexity [6].

In modern days, manufacturing becomes an integrated concept from factory floor operations to production activities and business level [7]. Recent advancements in sensing, actuation, and communication technologies have enabled enterprises to acquire, transmit, store, and process real-time data for characterizing factory behavior [8]. There are numerous control systems available to acquire the massive industrial data, including distributed control systems (DCS), supervisory control and data acquisition (SCADA) systems, and programmable logic controllers (PLC) [9]. The *Xively* middleware platform has a time series component that supports services for the querying and collection of data from smart devices [10]. The platform depends on representational state transfer (REST) principle for information receiving and service creation and data is organized into data points, feeds, and streams. Another middleware solution *RestThing* contains a data collection component from embedded devices and a database system to handle the storage of data sources and differentiation of their characteristics [11]. The storage of a variety of data was based on JSON and XML files. Mervat et al. [12] proposed an IoT data management framework and its components having data and source layers as a central part. Their architecture highlights the requirements for cross-layered and two-way design approach that address both archival and real-time data acquisition, process, and storage needs. The IoT middleware is a common architecture which supports the parallel and simultaneous applications and can effectively manage different heterogeneous data collection systems [13]. The middleware system acquires data using specific formats and storage capability. A recently introduced IoT data management framework, called distributed data service (DDS), well performed in acquiring, aggregating, and retrieving data from IoT middleware systems [14]. The DDS can process the massive data coming from various sources by a specific communication protocol and creating metadata modules.

However, all available data acquisition frameworks are focusing on specific issues but acquiring and storing the huge industrial data is still a challenging task. Furthermore, data coming from thousands of factory devices distributed in vast shop floors can arise in a high variety, different formats, and massive volume [15]. The recent literature shows that industries are shifting towards digitalization to increase their productivity, competitiveness, and performance [16]. Additionally, there should be further investigations of data acquisition techniques to address the existing and upcoming challenges. Some of the basic challenges are: (i) Data heterogeneity and dynamicity. (ii) Data visualization for diverse resources. (iii) Storing industrial big data. (iv) Standardization of communication protocols. (v) High productivity and product quality. (vi) Reliability and scalability of production lines. So, there is a need for a comprehensive data management framework that can efficiently acquire, process, and store the massive data through numerous physical and virtual factory devices to fulfill the manufacturing objectives, as well as control the production lines.

Therefore, the major contributions of this paper can be summarized as follows:

- A framework of an IoT-based industrial data management system (IDMS) is introduced to efficiently collect and analyze the raw industrial data and urgent events by applying state-of-the-art communication protocols.
- The framework is comprised of five layers like physical, network, middleware, database, and application layer. Specifically, the middleware layer consists of various components, including resource management, event management, data management, and recovery management to provide a service-oriented architecture (SOA) for the end users and applications.
- A distributed database server is designed to control the heavy data traffic and avoid communication delays. The raw industrial data is aggregated and converted into a structured database.
- The structured data is delivered to the cloud server for permanent storage, where different data mining and machine learning algorithms can be executed for knowledge extraction.

Consequently, experimental results from a smart factory case study illustrate that the proposed IDMS can efficiently acquire the massive data, and successfully monitor the production line activities on the shop floor. Moreover, it will advance the factory automation processes by improving assets utilization and increasing the speed of time to market. Note that our framework only focuses on management of industrial data and offers distributed storage server to store this data before transmission to cloud server. This would help to achieve smart manufacturing goals such as high resources awareness, high flexibility, and high productivity.

The rest of this paper is organized as follows. Section 2 provides a brief analysis of the related work. Industrial data characteristics and lifecycle within an IoT environment are discussed in Section 3. In Section 4, industrial data management framework is proposed, and its various components are described for the acquisition of heterogeneous data from factory floors. Section 5 presents the results from a case study to validate the proposed framework. Finally, Section 6 summarizes the conclusion.

## 2. Background and Related Work

Many efforts have been made for the IoT data acquisition, processing, and storage, especially in the last two decades [17]. The major objective is to efficiently acquire data produced by heterogeneous and distributed devices, aggregate this data, and store it for later use. For this purpose, IoT middleware systems provide solutions by implementing data acquisition frameworks. These solutions are very diverse in their programming abstraction level, design approach, and implementation domain.

### 2.1. IIoT Scope

The market associative and industrial experts with the IIoT promises tremendous growth and they are predicting a lot of saving cost, high productivity, and estimated budget to reach $123.89 Billion by 2021 [18]. Connecting tens of billions of industrial devices will revolutionize businesses by increasing process efficiency, reliability, and safety. Some companies like Honeywell, Cisco, Emerson, Uptake, Yokogawa, and GE are providing fully integrated smart factories systems using IIoT platforms, but their scope is to cover large-scale activities like railway logistics, power generation and distribution, oil and gas operations, and aviation fleet [19]. A fascinating report describes the achievements of industrial data analytics based on advances in the areas of machine learning, IoT, and industrial data analytics (IDA) [20]. According to the report, the IIoT and IDA have increased revenue, customer satisfaction, product quality of the smart factories up to 33.1%, 22.1%, and 11.0%, respectively. Moreover, the report says that 68% of manufacturing companies have their local IDA strategy, 46% have some potential organizational platforms, and only 30% have accomplished actual IDA projects. That means 70% of the companies still don't have standard data acquisition and analytics platform to make their business more intelligent and profitable.

Although many industries are already using the IIoT extensively but still there is a lot of space to do more work in this field. According to a survey conducted by PwC in [21], only one-third of manufacturers in the US is acquiring and using data generated by smart devices to improve their operating and manufacturing processes. The IIoT is contributing in almost all areas of modern manufacturing processes including supply chain, process control, logistics, maintenance and infrastructure [22]. A research study provided by Heymann in [23], predicted the transformation cost of current industries into smart industries (Industry 4.0) may be up to 267 Billion Euros of the German economy by 2025. However, adopting IIoT as a smart manufacturing solution can face some of the tremendous security and data privacy issues. Jayasri et al. [24] discussed major challenging issues of IIoT such as (i) data and service security, (ii) trust, information privacy, and data integrity, (iii) scalability, and (iv) interoperability.

### 2.2. Data Acquisition Frameworks

A huge volume of data is being generated by millions of IoT devices, and periodically reporting some abnormal or certain events [25]. After acquiring and processing, this data contributes to the

optimizing of industrial environment, online manufacturing, and monitoring systems [26]. An IoT-based real-time monitoring system for steel casting industry is presented in [27], which integrates different data processing approaches including data conversion, protocol conversion, and data filtering. The focus of their study is to address the data heterogeneity and communication protocol multiplicity challenges in the continuous industrial environment. Wang et al. [28] suggested a supervisory controller based on radio frequency identification (RFID) technology for shop floor production system, which collects data through RFID tags and readers. Communication of data is realized by industrial field bus and PLC technologies, and host computer implements the production process control and material handling devices. A robust, simple, intelligent, and cost-effective industrial control and data acquisition system was highlighted in [29]. The proposed system was in correlation with DCS, SCADA, and fully integrated automation, that means manufacture procedures and optimize processes are done at the same time. Razzaque et al. [13] conducted a survey to discuss various proposals of IoT middleware. Most of these proposals addressed wireless sensor networks (WSN) as a key component of IoT, but not mentioned other core elements like; SCADA, RFID, and machine-to-machine (M2M). They presented a comprehensive analysis of available IoT middleware systems and proposed a set of functional, non-functional, and architectural requirements for a potential IoT middleware solution. They also highlighted some challenges, open research issues, and future research directions in this area. Anjomshoa et al. [30] proposed a Social Internet of Things (SIoT) framework to acquire behavioral data from users' mobile sensors and social network applications. Different features were extracted to monitor the social behavior of individual users using different ML algorithms like support vector machines (SVM) and density-based clustering of applications with noise (DBSCAN).

Some of the basic challenges (i.e., scale, unknown topology, deep heterogeneity, inaccurate metadata, unknown data-point availability, and conflict resolution) of a service-oriented IoT architecture were focused in [31]. The study proposed a middleware for data composition and service discovery by applying a probabilistic approach to meet these challenges. The proposed solution consists of three basic modules such as discovery, estimation & composition, and knowledge base. Fremantle et al. [32] presented a Web Services Oxygenated (WSO2) middleware having a monitoring module called *Business Activity*. This module was responsible for acquiring data streams from the physical environment and processing them in real-time, predictive, and interactive modes. The WSO2 was also able to collect data from any data source of JavaScript clients, Java agents, and concerning events. It published Java events using an API for batch, combined processing, and used distributed database storage such as MongoDB or Casandra. A cloud-enabled smart manufacturing prediction system was presented in [33], which defined that data acquired by the manufacturing hierarchy can be used to improve productivity, efficiency, and profitability of smart factories. A variety of cloud platforms and initiatives are offering numerous service platforms, such as Platform as a Service (PaaS), Infrastructure as a Service (IaaS), and Software as a Service (SaaS) to the manufacturers. Authors also discussed some basic challenges and research directions about the data collection and management for cloud-enabled prediction systems. Iqbal et al. [34] proposed a data analytics framework using context-aware at fog layer to acquire, process, and analyze the real-time, near real-time, and batch services data from an Internet of Vehicles (IoV) environment. The proposed framework enables even in high mobility system and provides novel applications such as safety, emergency traffic plan, and adaptive signal scheduling. Some latest industrial communication standards (i.e., MQTT, XMPP, AMQP, DDS, and OPC-UA) and their comparison were presented in [35]. Although all these protocols were very different and even, they didn't fill comparable role strictly, but DDS and OPC Unified Architecture (OPC-UA) were well designated as they had much more features than for just sending the bits.

Secure data transmission is also a major concern in an industrial environment. Aloqaily et al. [36] proposed an instruction detection model for safe transmission of data between connected vehicles in a smart city. This model assures the transmission of data only between trusted third parties, cluster-heads, and service providers. They used a hybrid-based detection system based on D2H-IDS to reduce the dimensionality of data and ID3 based decision tree (DT) to classify the attacks.

In [37], the authors suggested a comprehensive list of privacy and security guidelines for communication and edge-nodes level to develop a secure IoT platform. Two emerging technologies such as blockchain and software defined networking (SDN) were illustrated to discuss the IoT security concerns. They also investigated all possible threats and attacks at all levels elaborated various security goals such as integrity, privacy, confidentiality. A hybrid-intrusion detection system (HIDS) has proposed in [38] for anomaly detection and signature detection from sensors' data. The authors implemented machine learning algorithms called random forest (RF) to identify known intrusive behavior and E-DBSCAN to recognize unknown intrusive behavior. The proposed HIDS architecture applies a hierarchical trust-based combination of sensed data which passes through either anomaly detection system or signature detection system.

## 3. Industrial Big Data

The technological advancements, especially in ICT, automation, and production, are increasing the number of internets connected devices. In the presence of these technologies, machines can communicate with each other as well as with the end products by using the IoT applications [39]. The factory devices from shop floors are continuously generating a massive volume of data [40]. The Cisco IBSG predicted that there would be 25 and 50 billion connected devices ranging from smartwatches, smartphones, ATMs, and PCs, to shipping containers and smart factories, by 2015 and 2020, respectively [41]. We represented the ratio of internet-connected devices over time, as shown in Figure 1. The data generated from these devices would be complex, scalable, and huge in volume. The data can be in the form of key-value sets or image/audio/video contents which are usually geo-stamped and time-stamped. So, this massive amount of data can be defined as Big Data and the IoT environment as a source of Big Data [42]. The major costly projects in the smart factory's era are related to data acquisition, aggregation, and analysis; which contain the total cost of 21%, 17%, and 14%, respectively [20]. In modern days smart factories era, the data generated by industrial devices have reached to more than 1000 EB of total volume annually and will increase on regular bases [43,44].

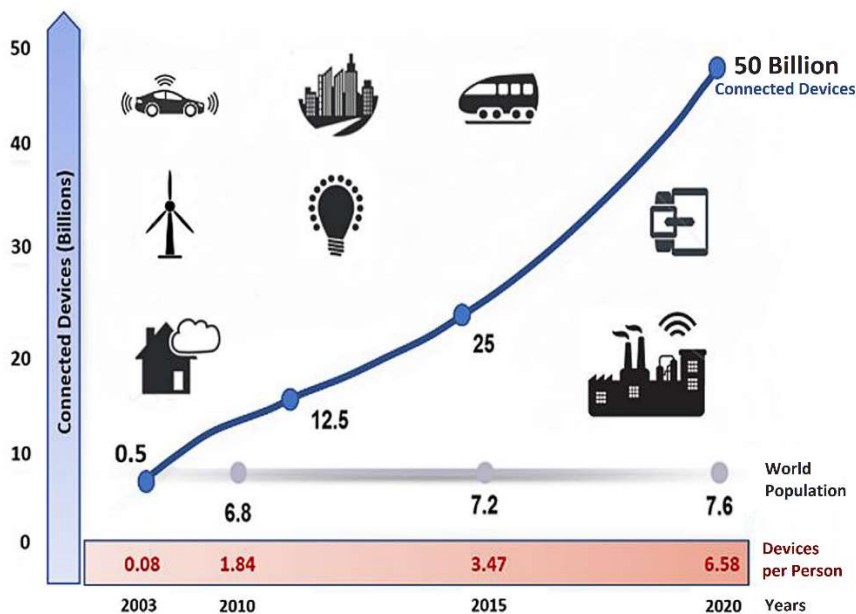

**Figure 1.** Future evolution of IoT devices.

*3.1. Characteristics of Industrial Data*

The IIoT is considered as a revolution that has completely changed the industrial face in a profound manner and transformed the traditional manufacturing system into a digital ecosystem. Innovations in information and operational technologies are enabling huge data exchange between factory devices, and thus improve the enterprise performance, responsiveness, and flexibility. As

factory devices involve mobility and can capture real-time events, so they can generate a massive amount of diverse data at very high speed. Smart factories have created a new term called Industrial Big Data, where the variety, volume, and velocity of generated data is reporting at very high rates [45]. Industries can also exploit the collected data to create businesses advantages and improve their competitiveness by predictive analysis [46]. Moreover, extraction of useful information from industrial data is a challenging task.

It has been proved that smart machines are more accurate than human beings for communicating and consistently capturing data from a distributed platform of IIoT [1]. This data allows industries to save precious time and money by picking up problems sooner and support business efforts. All available business management and IT tools have been interconnected to the network via latest communication standards. These smart enterprises know how to collect, transmit, and analyze huge volume of industrial data. The IIoT has a potential for supply chain efficiency, traceability, and quality controls, especially for manufacturing systems. Data produced directly from human operators and machine tools is very important because it provides valuable information to the manufacturers for making these tools more healthy, scalable, flexible, and adaptive. Mostly, available IoT data management systems focus on the collection of data promptly for making early and intelligent decisions, but with a limited capacity of permanent storage for later usage. The industrial environment also contains a variety of data resources such as embedded and smart, archival and real-time, and mobile and stationary. So, all these circumstances show that there should be a comprehensive data management tool to support such a scalable industrial data.

## 3.2. Industrial Data Lifecycle

Industrial data is a very important resource that can be more critical for worldwide manufacturing business processes and the source of huge wealth if handled properly. Managing this data requires high processing and storage capabilities due to its huge, complex, and unstructured nature. The lifecycle of industrial data can be defined with the help of three phases like physical, middleware, and application as illustrated in Figure 2. The whole industrial environment has been divided into two sub-sections like the real and digital world. In real-world environments, the raw industrial data with various data types, formats, and diverse dimensions are generated by numerous physical components of smart factories. The physical devices component contains all valid data sources such as sensors, web-generated data, databases, and third-party applications. This component is also known as data discovery. After implementing the digitalization and aggregation processes this data becomes a part of the digital-world in binary form, where middleware and application components offer numerous services to manage it. The middleware component addresses interoperability across various factory devices, device discovery, scalability, management of massive data, context awareness, and the security features of the IoT environment [47].

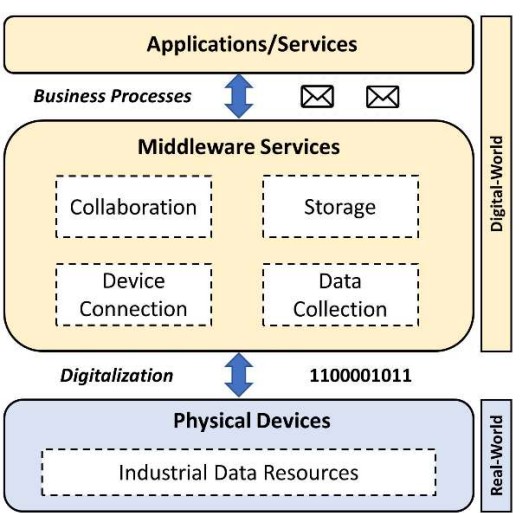

**Figure 2.** Industrial data lifecycle.

In the middleware phase, the data collection module directly interacts with the physical devices and exchange binary data using commands/response method. The data is directly or indirectly acquired according to scientific and business demands from all valid sources. After applying filtering, sorting, aggregation, and classification techniques, the processed data is sent to the storage module for permanently storing into available repositories for later use. Recently, these repositories use cloud computing systems to store industrial data and provide quality attributes such as reliability, availability, security, scalability, and robustness [40]. All factory devices have been deployed with the IIoT system through the device connection module. This module responds to regular and event data streams accordingly. For example, all pre-processing functionalities are applied first for regular data stream (offline data) and then stored in permanent repositories, whereas specific events (online data) are directly transmitted to the application server for a quick response. The collaboration module provides services to the consumer of data for direct access to the physical devices individually. By doing this, a user can find the metadata about devices (i.e., type, location, timestamp). The application component interacts with the middleware by receiving request messages and sending responses to those requests. This phase also provides the platform for programmers and analysts to make the available data useful and generate new opportunities and services. The data is examined with the goal to optimize processes, extract useful insights, and make better and early decisions.

## 4. Proposed Framework

Compared with traditional manufacturing processes, smart manufacturing has characteristics such as deep integration, huge volume of data, and high correlation. Accordingly, most of the manufacturer are still facing various challenges to acquire industrial data. So, this section presents a framework of Industrial Data Management System (IDMS), for real-time and scalable data collection, transmission, processing, and storage. Different layers of IDMS framework with their corresponding modules are shown in Figure 3. The IDMS allows the manufacturers to acquire raw industrial data from the factory floor and partially store into local repositories before streaming to the cloud server. The proposed framework consists of five basic layers where each layer having different functional components. For example, physical layer contains all industrial sensors, actuator, and field devices, which are responsible for the creation of raw data and unique events. The communication layer implements latest industrial protocols and makes sure of the secure connection between each layer of the system. This layer also controls streaming of data, queries, requests, and results. The middleware layer consists of different functional components to provide support for discovery of diverse data sources and apply data processing processes. The database layer supported by the local repositories provides partial storage for distributed industrial data to overcome communication delays and intensive workload on the cloud server. The application layer handles the quires from consumers and provides real-time data analysis. Each layer of the proposed framework is explained in the following sections.

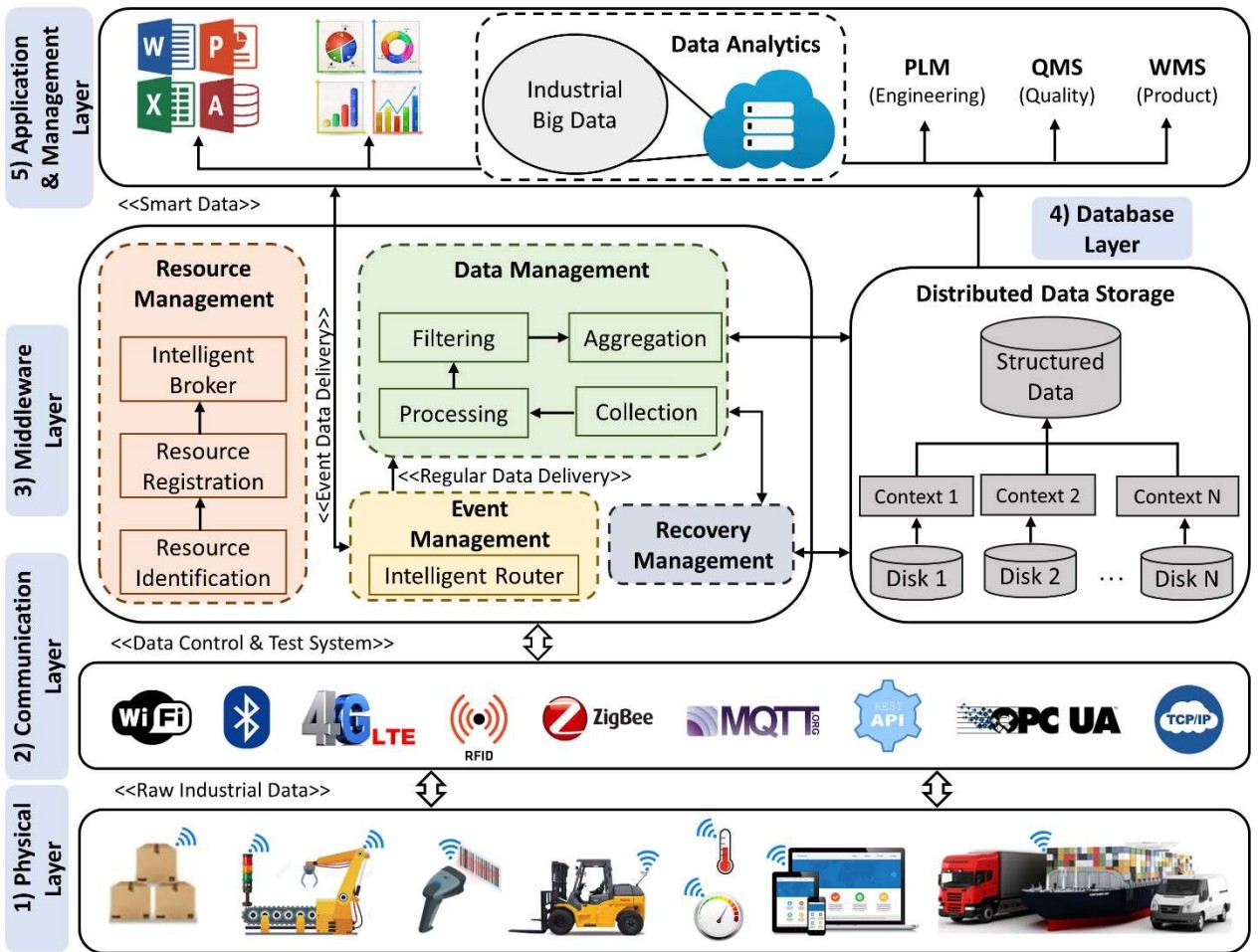

**Figure 3.** A framework of an IoT-based industrial data management system. **Note:** PLM indicates Product Lifecycle Management, QMS Quality Management System, WMS Warehouse Management System.

## 4.1. Physical Layer

The physical layer contains all data producer entities of the manufacturing system and modulus. These entities can be flow meters, servo meters, robots, conveyor belts, PLC's, machine visions, smart containers, embedded chips, and various devices on the shop floor. A real-time data is acquired from factory floor and transmitted to the upper layer by various adapters using state-of-the-art industrial communication protocol called Open Platform Communication Unified Architecture (OPC-UA). These adapters have been deployed with different sensors such as temperature, pressure, energy, vibration, rotational speed, force, torque, and acoustic emission to monitor the real status of machines in digital form. All physical devices on the shop floor are integrated with each other and every device has unique identification so that any data consumer can access them individually. We preferred location-based identification as it is more efficient for scalable data acquisition within a distributed environment. Local aggregation has been applied at this layer to minimize the storage and transmission cost of the raw industrial data. The local aggregation modules acquired data from data sources and summarized this by discarding the less important or homogeneous data streams. This also helps to transmit data efficiently with minimum delay and makes sure the real-time data acquisition system.

## 4.2. Communication Layer

The communication layer provides human machines interface, connects all layers of the proposed framework, and offers transmission links between data producers and consumers. This layer also handles communication between distributed factory devices within a vast industrial area to concentrate data collection, processing, visualization, and storage. Normally, the size of industrial devices is kept small, which decrease their computational and processing capabilities. Moreover, most of these devices are powered by batteries, so energy management is also a critical issue for them. To overcome these issues, a wireless sensor networks (WSN) is implemented for monitoring of industrial devices. The WSN offers scalability and flexibility, and it has ability to work with many devices collaboratively to achieve the common goals [48].

The industrial data should be efficiently collected from the physical layer and transmitted it with high throughput and low latency rate to upper layers for further processes and analysis. For this purpose, the communication layer provides some solutions as a central hub. Various communication technologies such as RFID, Wi-Fi, Bluetooth, Wi-Fi direct, 4G LTE, Z-wave, ZigBee, etc. are being used to transfer heavy data traffic with latency guarantee and high bandwidth support. These technologies have ability to process and store data at a small level for a considerable time duration. This layer assures the data security and employees privacy by protecting the framework from unauthorized access and makes sure of safe transmission of data on every phase of the data lifecycle. Some common protocols (i.e., IPv6, MQTT, SOAP, REST API, OPC-UA) have been suggested for this purpose. The IPv6 protocol offers an unlimited number of IP addresses and provides a platform for symmetric and bidirectional connection of billions of smart devices globally. The MQTT is used to acquire data from various devices and transmit to the middleware layer by targeting the large industrial environment that needs to be controlled and monitored by the cloud server. This protocol works on the top of TCP, so that system does not lose any data stream irrespective of how long repeats take. The REST API protocol is specifically used for secure collection of data from IIoT. Data is collected in formal message arrays, and receivers split these into individual message packets and identify the device. After having authentication, the packets are sent to the message queue for pre-processing tasks.

OPC-UA has been defined as a standardized industrial communication protocol for the reliable, secure, and vendor-neutral transmission of raw data from the production planning and manufacturing level sensors [49]. It provides required information to every authorized person and application at anywhere and anytime. Numerous key features such as scalability, platform and programming language independence, internet capability, and high availability are offered by OPC-UA. An information model and integrated address space are defined to represent the processed data, historical data, program calls, and alarms. Embedded OPC-UA adapters is deployed at geographically distributed factory devices and their enterprise resource planning (ERP) systems can be directly connected with each other by firewalls. Moreover, OPC-UA is tightly integrated with fundamental security requirements. Its security mechanism depends on detailed analysis of various security threats. It provides three level security architecture such as user-level security, application-level security, and transport-level security [50].

### 4.3. Middleware Layer

A vast network of interconnected factory devices, a substantial number of events generated by shop floor machines, and complex IoT technologies meet new challenges for development of IIoT applications. In this context, a middleware layer offers various services for applications development by integrating heterogeneous computing and communication devices. This layer supports interoperability within the diverse applications and services running on these devices. Many operating systems have been developed to support the IoT middleware solutions, but limited systems are available for manufacturing environments [51]. Recently, middleware solutions are gaining more importance due to the key role in simplifying the growth of new services and applications [52]. In our proposed solution, the middleware layer acquires data from the physical components of the IIoT system and effectively manages all data resources. This layer directly interacts with data producer at the physical layer and data consumer at either the application layer or distributed data storage layer.

It also handles the requests and responses between these layers. Data is managed by creating novel methods of acquiring, transferring, and storing at this layer. Various challenges are resolved that contain indexing, querying transaction, and process handling. In the proposed framework, middleware layer has various components for converting the raw industrial data into processed data and transmits this data to distributed data storage for later use. Each component has been further elaborated in the followings.

4.3.1. Resource Management

The resource management component makes sure of the discovery and arrangement of data sources by storing the database fragment locations for updating and querying purposes. There are some basic issues for industrial data sources that make this system more challenging for real-time queries execution. First, data generated by the factory devices is heterogeneous and scalable due to continues participation of new data sources. Second, the location of data sources become diverse due to vast and mobile nature of the manufacturing objects. Third, there are various unique data sources with no defined metadata rules. To meet all these issues, there should be a layer on top of the physical layer to manage the unification and identification of data sources for processing of queries. Resource management offers some solutions for this purpose, as described below:

- *Resource Identification:* The industrial environment is flexible and can easily accommodate new sources and sub-systems when necessary. So, the resource identification mechanism handles the continues and real-time queries for these sources. Identity of new sources is verified by authorizing and passing the device specification to the metadata store.
- *Resource Registration:* This module is responsible for data source registry and provides functions between physical and application layers to make agreements for sharing of data and queries. Its major concern is the query execution plan by handling the queries from a data consumer to the result of these queries from data producer. By doing this, the registration module becomes aware of the locations of each data source and every data item becomes uniquely identifiable.
- *Intelligent Broker:* The broker behaves as a mediator between data consumers and data producers. It supervises and keeps records of each data exchange transaction and provides the rollback in case of incomplete and faulty data exchange transactions. Data analysis and quality related services are also provided by the broker.

4.3.2. Event Management

After collecting the raw industrial data through smart devices and the intelligent network, now it is delivered either directly to the application layer for final responses of urgent events or to the data management component for pre-processing and storage of normal data stream. We have implemented an intelligent router by setting define thresholds to detects specific events for certain conditions. If an urgent event was generated by shop floor machines, the router activates the event operation to respond to that event simultaneously. The router is smart enough to differentiate the regular and event data streams. When the event disappears and it was successfully responded, the router will stop the event operation and activate the regular data operation. Events are very crucial for online monitoring systems, so this module plays a key role by detecting them from heavy data traffic. This process also makes sure of the early diagnosis of failure and improves the lifetime of machines.

4.3.3. Data Management

Data generated by factory devices are delivered to the data management component for pre-processing. We followed the data distribution system (DDS) technique for data management, proposed by Huacarpuma et al. [14]. The DDS can support multiple connections within IoT environment simultaneously. It comprises two main functions: data collection and aggregation. During the data collection function, data is acquired from devices at the physical layer and applied

initial pre-processing steps. Furthermore, this data is sent to the aggregation component for summarizing into different chunks. Finally, the partially structured and aggregated data is sent to the distributed data storage module where it is converted into the fully structured format by applying context extraction techniques. Various modules of the data management component are shown in Figure 4 and further explained in the followings.

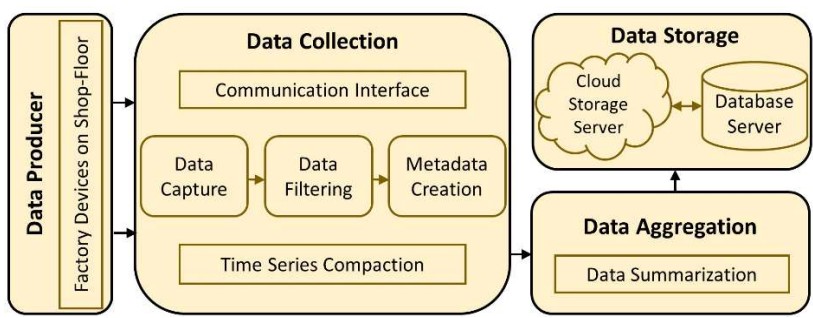

**Figure 4.** Data flow at data management component.

Data Collection

This component collects industrial data in a better way and applies filtering, reasoning, and fusing on it. To achieve this task, sensory data need to be acquired in a location-sensitive and timely manner. Data collection component is responsible for the collection of raw industrial data and pre-processed this data. It consists of five modules:

- Data capture: Acquires the raw data from various sensors using *Apache Kafka* architecture which supports distributed, open source, and messaging system [53]. Moreover, data is stored without any processing and can be removed with retention time accordingly. This module supports diverse data sources and performs several processes simultaneously.
- Data filtering: Verifies the domain of collected data by the previous module. Filtering requires querying the database and applies filtering rules. Less important and out of context data is discarded using *Apache Storm* system [54], which is a distributed, real-time, and open source computation system. Thus, it minimizes the storage cost and makes sure of fast data transmission.
- Metadata creation: Some important metadata objects are obtained, like data type, measuring units, time stamp, and geolocation. This module also describes the specific industrial environment, data, and applications.
- Communication interface: Communication between each module of the data collection component is organized. Various types of data are translated into a single format so that system can understand. For example, the data coming from various devices with different formats are translated into *JSON* massage structure first, and then sent it to next phase for data aggregation.
- Time series compaction: Collected data is organized into diverse groups and sorted these groups according to the certain time windows. These groups can be ordered according to speed and volume of data production.

Data Aggregation

The pre-processed data is transmitted to the aggregation component for further summarization. The aggregated data is more significant than the raw data collected by factory devices. The data stream coming from the physical layer is separated into data summarization modules as described below.

- Data summarization: The datasets of various devices are represented into groups according to time-period. It reduces computational and storage cost and improves consultation performance by minimizing the volume of data. So, the event table generated by the data collection

component is now changed into an aggregated table that shows the data values into per-minute, per-hour, and per-day format. Each format has different numbers of attributes according to data size such as per-hour format consists of six attributes like device_id, date, per_hr (hour), count (number of readings in an hour), context (latest reading place), and total (size of value in an hour).

### 4.3.4. Recovery Management

Recovery management module plays a vital role when the database server goes down or crash suddenly. This module detects the problem from link failure by maintaining numerous communication links between the middleware and distributed data storage layers. There are several solutions to overcome this specific problem, one of those is the adaptation of checkpoint that makes the content of database periodically saved [31]. All latest dumps are stored during the recovery period of the system. The problem is solved by accessing undo-redo list to store again all the transactions state until the last checkpoint. Remote backup is used for data storage during the recovery time of the server. This novelty method provides a sense of safety and security for real-time data storage. The backup is provided for every single bit of data simultaneously at two distant levels, directly connected to the server and other is sent to remote level as a backup. Whenever the server goes down, the backup system detects the failure and activate the remote storage. This process is so instant that even sometime user can't realize the actual failure.

Figure 5 illustrates a flowchart to describes the algorithm of data backup provided by the recovery management component. Start terminator shows that the proposed IDMS is being activated for acquisition of real-time data from shop floor. Preparation process indicates that data is being normally transmitted from data management component to distribute data storage component. Database server status decision checks the communication link between the middleware layer and database layer. If the server is properly working, then the data is transmitted normally to the database storage. Otherwise, new backup policy for data storage document will be activated to perform recovery of entire data and other records. Meanwhile, this document will notify the system about the sever failure or sudden damage of link. Execute the backup job process terminate backup policy when the server is recovered. Data is backed-up on hourly, daily, and weekly bases. When the database server is recovered from the failure, all the backed data is transmitted to the server and backup storage remains inactive till the next failure.

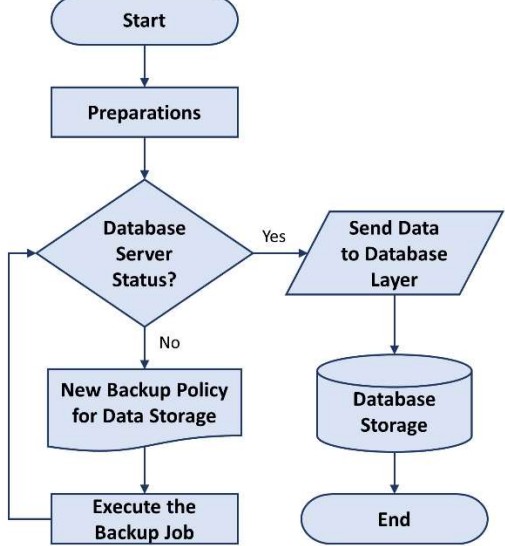

**Figure 5.** Flowchart for database recovery.

### 4.4. Database Layer

The aggregated data is stored into local repositories to enhance the capability of computation and storage. The entire process of changing the heterogeneous industrial data into the structured format is defined as follows.

Distributed Data Storage

A factory floor contains the chain of machines and each machine has numerous devices to generate continuous data. However, this data cannot be transmitted directly to the cloud due to heavy data traffic and limited network bandwidth. This may cause of transaction delaying problem. So, to avoid these delays the proposed framework offers a novel context-aware data storage disk for each machine on the factory floor to ensures the updates durability. By this, we can have online access to the real status of each machine individually and their performances can be compared. Afterword, we applied context extraction on the processed data to acquire the information about environmental variable such as geo-location of factory device and type of data generated. Contexts are useful to overcome the diverse resource problem and allow the use of environmental data to support for making the prompt decision when necessary. Contextual data such as *data-space*, *time*, and *location* can be used by various applications to make the recommendations.

Context extraction from large industrial data eases the process of communication, storage, and analysis by offering only relevant data for specific services and applications. The unnecessary data is trimmed to minimize utilization of resources due to limited transmission, computing, processing, and storage capabilities. The trimming process ensures avoidance of faulty, incomplete, and duplicate data. Faulty and incomplete data is requested again from the data management component; duplicated data is combined to get the most recent data. So, the available data becomes error-free, complete, clean, and contextual in nature. Each context has more meaningful information and they are combined to make a well-structured intelligent data. This intelligent data plays a vital role to define the health status of the manufacturing system and can improve the productivity, process quality, and efficiency by early faults detection and making intelligent maintenance decisions. Lastly, the structured data is transmitted to the cloud server for further analysis and processing by cloud-gateway. Before final transmission, the data is organized into the small packets and sent to the application layer using REST-API protocol.

*4.5. Application & Management Layer*

Application services and management issues are managed at this layer. It provides security and easy access to different data storage services. Industrial standardized protocols are implemented for privacy and protection of the data and the end users of the system. It is ensured that all the running applications do not hold any malware and providing accurate and clear information to the consumers. Moreover, data consumers have been provided full access to real-time data gathered by factory devices. According to the nature of data, various security applications have been implemented using OPC-UA security mechanisms [50]. The OPC-UA security measures deal with authentication of users and applications, confidentiality and integrity of the exchanged information, and the justification of function profiles. The proposed security concept consists of three basic levels: user-level security, application-level security, and transport-level security. The user-level security mechanisms provide access to specific users and their role while setting up new sessions. Application-level security is a part of the communication session and contains the exchange of digitally signed certificates to authenticate an application during secure channel creation. Transport-level security is used for encryption and signing each data entity during communication sessions. Encryption avoids eavesdropping, while signing certifies the data authenticity and integrity.

The application layer provides a platform to extract useful patterns from industrial data and convert it into knowledge that is used for the future improvement, early and better decision making, proper functioning, and novel business opportunities. Information produced by industrial data is delivered as the set of services to the end users such as Product Lifecycle Management (PLM), Enterprise Resource Planning (ERP), Supply Chain Management (SCM), Manufacturing Execution

System (MES), Quality Management System (QMS), and Warehouse Management System (WMS). Various machine learning (ML) and deep learning (DL) techniques such as Artificial Neural Network, Random Forest, Support Vector Machine, Logistic Regression, Recurrent Neural Network, Restricted Boltzmann Machine, Auto Encoder, and Convolutional Neural Network can be applied on the industrial data for real-time and the multitude IIoT applications [55]. Moreover, the DL models got outstanding results in different applications of natural language processing, text recognition, image recognition and games. With high-volume modelling and automatic feature extraction abilities, DL offers an innovative analytics tool for intelligent manufacturing especially in the big data era. This layer also contains data analytics component as explained below.

Data Analytics

Data analytics is responsible for converting context-aware data into intelligent data. This intelligence is appreciated by analytics and can provide both delay-tolerant and delay-sensitive applications at shop floor and middleware layer. The open cloud storage server is configured to support real-time queries and data coming from the IIoT middleware. The cloud computing techniques enable the storage of massive data from factory devices, provide ubiquitous access to useful information for decisions making and collaboration among various industrial tools [56]. For this purpose, we conducted online prognosis and diagnosis of factory machines and monitoring processes by implementing cloud-based ML and DL techniques. These techniques can be implemented to extract useful features from acquired industrial data to check the real status of individual machine on the shop floor [30]. Numerous ML and DL models such as classification, clustering, and regression need two types of datasets such as training dataset to train the algorithm for prediction and test dataset to evaluate the algorithm. Once the model is evaluated, it is used for prediction of machine health, earlier failure detection, and shortage of stock to make sure of smooth manufacturing processes. All cloud service providers are using common standards so that their consumers should have a choice to use a specific provider according to the system demands. For example, various cloud platforms have been proposed to offer a variety of services such as PaaS, IaaS, and SaaS to manufacturers. Many interoperability standards have also been suggested for integration of data such as MTConnect and OPC-UA [33]. Software and hardware vendors are providing cloud-enabled prognosis and diagnosis solutions, such as remote monitoring of factory devices and machine tools.

The IIoT applications provide on-demand services through a cloud server which extends beyond the need to store sensors data in a scalable manner. As the industrial environment is geo-distributed and require mobility support for low latency and location awareness, so we introduced the use of *Micro-Cloud* or *Fog* techniques. These techniques enable new services of data analytics and provide real-time data computation and storage. Afterward, the cloud server can be supported by MongoDB due to its NoSQL database feature for the storage of industrial data. We preferred REST-API protocol for the transmission of data on the cloud server as it is more reliable and provides support for a distributed industrial environment [52]. It uses local hardware and software gateways that act as intermediate controllers. Locally encryption is provided to enhance the security and process the data locally for minimizing the size of data sending to the cloud server. So, it helps to criticize the chances of end-user privacy violations which can happen during data acquisition and transmissions. Some important features such as *POST*, *GET*, *PUT*, and *DELETE* are provided so that producers and consumers of data have equal opportunities to use it.

## 5. Results & Analysis

To evaluate the performance of the proposed framework, this section presents result analysis of a real-time factory automation case study. Due to IoT applications, industries are relying on the network of interconnected physical objects to connect devices, places, and people. Distributions of intelligence in the field allows the connected devices to publish their data in a standardized format. Smart brokers make this data available in a transparent way to the end users. This approach is helpful

to identify the location of data sources without custom programming. For example, small and easily deployed RFID tags are attached to industrial products to find the exact location, so that these should be easily accessible by the plant manager, shift supervisor, and assembly worker.

Meanwhile, the proposed framework supports wireless network structure such as WSN to provide a high-performance and scalable networking platform for strong connections to employees and end users. Moreover, manufacturers at ERP level can take advantage of this facility by mobile accessing to production line data from floor manager's smartphones and PCs. This advanced level of connectivity helps to keep production manager and top floor management up-to-date and makes sure of high-quality products and timely delivery. Floor managers are continuously aware of all production lines by deploying RFID tags. They know when the production needs to be slow or fast to meet hourly, daily, and weekly targets, and how efficiently workers are finalizing their respective production phases. With such great visibility on production operations, manufacturers get a better understanding to meet challenges and achieve higher efficiency.

### 5.1. Case Study: Water Treatment Plant

Here we have a *Grundfos-CR* smart pumps for the sake of water treatment, filtration, washing, and cleaning system as a case study that are the part of a factory floor. We implemented the IDMS framework to acquire real-time data from various sensors deployed on these pumps. The main objectives of this case study are to find the volume of generated data, measurement of energy consumption, and early prediction of potential failure. We deployed four sensors on each pump such as flow meter, current transducer, accelerometer, and power meter to measure the water flow rate, energy usage, noise and vibration, and power of the pump, respectively. The real-time data was acquired and delivered to the cloud gateway using OPC-UA industrial standard, where we calculated the size of generated data. All sensors with their complete features are listed in Table 1. Total volume from a single pump was calculated after measuring the size of each sensor data, and the result shows how much size of data can be produced by these sensors. Just imagine the size of data if there are 10 similar pumps working in different shop floors of the same factory or if we calculate the volume of data from five similar factories with total 50 pumps.

**Table 1.** Devices deployed with the smart pump.

| Device Name | Reading Value | Value Range | Data Produced (MB/h) |
| --- | --- | --- | --- |
| Flow meter | Water flow rate | 50 [gmp] | 43.20 |
| Current transducer | Electric current | 30 [kA] | 43.20 |
| Accelerometer | Machine vibration | 3-10 [kHz] | 43.20 |
| Power meter | Pump power | 3-25[hp] | 43.20 |

MB/h denotes megabyte per hour.

Figure 6 shows the data output from individual sensors of the smart pump. The data size calculation is depended upon the operations that proceed by reading sensors in every 10 seconds and transmitting the corresponding 120-byte message packets. This represents that the redundancy from the dataset has been removed, where each record associated with a unique observation of data chunk (i.e., every 10 second times stamp), and each attribute characterizes a unique measurement (i.e., liquid flow rate). This well-structured dataset offers consequent processing components with a more general format to perform the transformation. In this way, data acquisition and pre-processing efforts related to data analytics can be highly reduced.

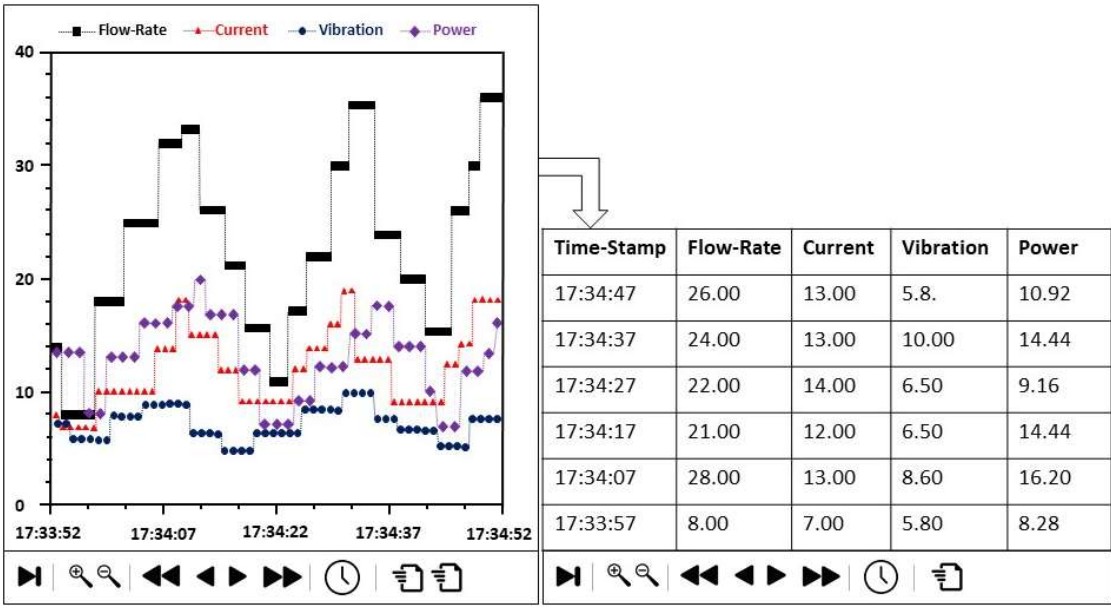

**Figure 6.** Collecting real-time data from various sensors of smart pump.

Figure 7 illustrates the volume of estimated generated data acquired by the IDMS framework for different numbers of pumps and for different time periods such as per-hour, per-day, per-week, and per-year. The process was done in increasing order of one to fifty pumps. The result shows that the generated data volume is proportional to the number of pumps being executed. Furthermore, data becomes huge when we measured it for a long-time duration. For example, data produced from 50 machines in an hour and during an entire year is 8.6 GB and 74,650 GB respectively. From these scenarios, it has been concluded that the proposed framework can be implemented in the vast industrial environment for the acquisition and storage of massive and heterogeneous data generated by factory devices.

After implementing ML and DL algorithms on the real-time collected data, we can check the exact status of the smart pumps, make intelligent decisions about maintenance schedule, and predict the machine failure by comparing the results with the historical data. We can also constantly track the real-time status of various devices from our home or office and determine if there are some defects. On behalf of the results, system will notify the field technicians and application manager about these defects, so that they can plan for early maintenance or replacement of the devices without waiting for massive failure to occurs. Predictive modeling of the system means it's really a better and more efficient solution for business processes based on having planned maintenance rather than unplanned maintenance for factory machines. When the machines are working normally and no event is being detected, then the technicians and application manager on the shop floor can determine the number of days left for next maintenance by early prediction from previous knowledge. So, by applying this framework, we can connect all the industrial devices with IoT and acquire real-time data. This data can be further converted into useful information by prediction modeling to increase the efficiency and lifetime of these devices by earlier maintenance.

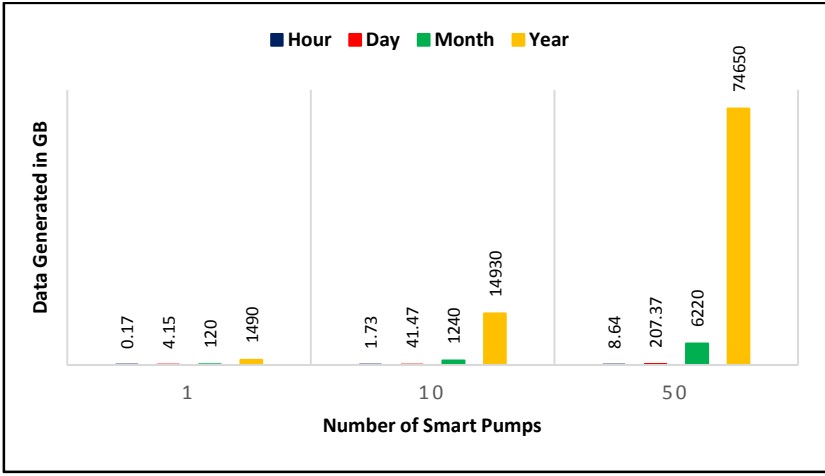

**Figure 7.** Data generation volume per-unit time for different number of pumps.

## 5.2. Real-Time Monitoring

Real-time monitoring is essential to find the production lines status and to drive smart decision in the manufacturing system. For this purpose, we presented an online monitoring system to validate the proposed IDMS framework for acquiring real-time industrial data, monitoring abnormal events, accessing historical data, and analyzing applications. The entire process for a single production line is shown in Figure 8. The automated production line requires changeovers, that resources be managed quickly and efficiently. The shop floor management wants to solve challenges such as production output, schedule transparency, and product quality updates. The IDMS offers the solution by integrating the information and operational technologies with its users to realize massive cost saving goals. It uses latest industrial communication protocol OPC-UA that offers high-performance, reliable, easy to use, and safe interfacing between production and high-level ERP and MES layers.

These days, manufacturers are relying on constant connections between industrial devices through the internet. So, Figure 8 shows that before final delivery on the production line, every product is deployed small size Wi-Fi enabled RFID tag by machine *A*, which provides exact status about the quality and location of every product to plant managers and assembly workers. Machine *B* receives the products and delivers them to machine *C* and *D*, which are responsible for the final delivery of products from the shop floor. Here every machine and product are communicating with each other and the monitoring system through a Wi-Fi router. The real-time data is efficiently collected using UPC-UA broker and after pre-processing, it is delivered to the cloud server for prognosis and predictivity analysis by MES. If there occur some abnormal or emergency events on the production line, such as any mechanical failure or some alarm, then the concerning threat is detected by event management module and directly delivered to top floor manager who is responsible to report back immediately and alert the floor technicians. In this way, manufacturers can easily access the line products data from anywhere and anytime. It will also help to keep the top floor management up-to-date by making sure the delivery of high-quality end products which were manufactured and distributed within time. For example, as the RFID tags are integrated with the PLCs of quality checking scale at the finishing point of the production line, high- and low-quality products results are directly transmitted to the MES level for the online monitoring of the production quality. This connection will also help the factory floor management to aware of the output of each production line, and to check whether they meet the production targets and whether the factory machines are working efficiently or not.

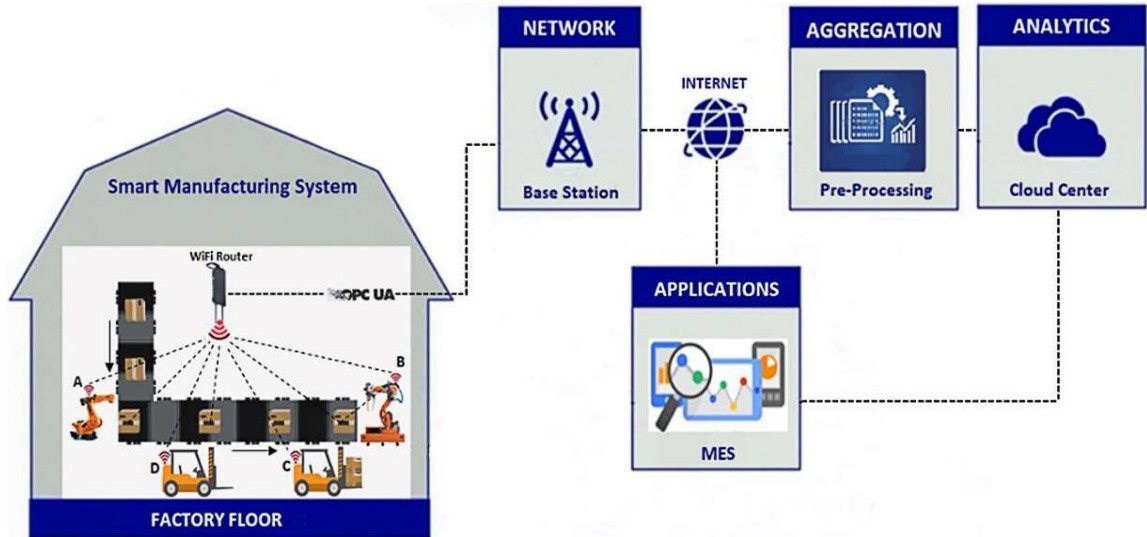

**Figure 8.** Online monitoring the operational processes of factory floor.

Thus, the key features of the proposed IDMS framework that make it distinct from others have been concluded as follows:

- It enables to efficiently acquire, process, and store the real-time, scalable, and heterogeneous data from a distributed industrial environment.
- Fast and secure data transmission is guaranteed by implementing state-of-the-art industrial communication protocols i.e., REST API and OPC-UA.
- Distributed data storage is provided to overcome the mobile and diverse data source problems.
- By integrating the fleet-wide information and sensor's data, the data volume is reduced, and useful patterns are identified to get hidden knowledge from industrial big data.
- On behalf of the extracted knowledge, smart decision can be made to decrease the downtime of factory machines and improve their health.
- Improves the efficiency and reliability of the manufacturing industries by reducing the labor cost, energy cost, and optimize maintenance scheduling.

## 6. Conclusion

The Industrial IoT is a complex topic that includes aspects of information technology, operation technology, statistics, and engineering. So, we proposed a layered framework for industrial data management system with five basic layers like physical, network, middleware, database, and application layers. The different modules of middleware layer support the retrieving and collection of huge industrial data generated by thousands of factory devices on the shop floor and extract useful information by applying context-aware approach. A distributed data storage is offered to process the data coming from mobile and vast industrial environments by a definite communication channel and metadata modules. We applied state-of-the-art industrial communication protocol OPC-UA for safe and fast data transmission. The case study concluded that our framework effectively supported huge industrial data acquisition and collected a real-time data of 8.6 GB, 207.4 GB, 6220 GB, and 74,650 GB in an hour, day, month, and year, respectively. Thus, it increased the manufacturing business processes by improving the revenue, customer satisfaction, product quality up to 33.1%, 22.1%, and 11.0%, respectively. Therefore, this study improved the industrial production scale and drive the development towards the smart manufacturing for more secure, sustainable, and efficient business.

**Author Contributions:** Conceptualization, M.S. and M.P.; Methodology and Validation, M.S. and J.Y.L.; Formal Analysis, Y.S.; Investigation, M.P. and Y.S.; Resources, M.S.; Writing—Original Draft Preparation, M.S.; Writing—Review & Editing, M.S. and J.Y.L.; Visualization, Y.S.; Supervision, M.P. and J.Y.L.; Funding Acquisition, Y.S. and J.Y.L..

**Funding:** This research was supported by the KIAT (Korea Institute for Advancement of Technology) grant funded by the Korea Government (MOTIE: Ministry of Trade Industry and Energy) (No. N0002429). It was also supported by Basic Science Research Program through the National Research Foundation of Korea (NRF) funded by the Ministry of Education (2017R1D1A1A02018718). This work was supported by the Technology Development Program (S2605169) funded by the Ministry of SMEs and Startups (MSS, Korea).

**Conflicts of Interest**: The authors declare no conflict of interest.

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
