# Peer review of "Framework of an IoT-based Industrial Data Management for Smart Manufacturing"

_jsan, doi:10.3390/jsan8020025_

Round 1
Reviewer 1 Report
The authors presented the industrial data management framework based on IoT, which can manage huge industrial data, support online monitoring and control intelligent production. The structure contains five basic layers. The solution manages regular data and urgent events generated from various devices in a distributed system. The collected data is used to improve the efficiency and forecasting of production lines.
The authors stated that the solution effectively supports the enormous acquisition of industrial data and increases production processes through the early maintenance of machines before their actual failure. To what extent and to what extent did this improve the results of industrial production?
The authors wrote "After implementing ML algorithms on the real-time collected data, we can check the exact status of the smart pumps". Which algorithms and how implemented can be used?
On page 5,6,7,15 and 16 you need to correct references to references, currently "Error! Reference source not found. "I suggest to improve (standardize) text formatting, eg page 11, 17. From page 15 the page numbering rule has been changed.
Author Response
Response to the Reviewer-1 Comments
Ref. No.: jsan-481009
Title: Framework of an IoT-based Industrial Data Management for Smart Manufacturing
Point 1: The authors stated that the solution effectively supports the enormous acquisition of industrial data and increases production processes through the early maintenance of machines before their actual failure. To what extent and to what extent did this improve the results of industrial production?
Response: Thanks for mentioning the important points. A well-structured industrial data management system can increase revenue, customer satisfaction, product quality of the smart factories up to 33.1%, 22.1%, and 11.0%, respectively. All changes have been highlighted in yellow color on page 3.
Point 2: The authors wrote "After implementing ML algorithms on the real-time collected data, we can check the exact status of the smart pumps". Which algorithms and how implemented can be used?
Response: There are various machine learning (ML) and deep learning (DL) algorithms such as Artificial Neural Network, Random Forest, Support Vector Machine, Logistic Regression, Recurrent Neural Network, Restricted Boltzmann Machine, Auto Encoder, and Convolutional Neural Network that can be applied to analyze the data to check the status of smart pumps. All changes have been highlighted in yellow color on page 13.
Point 3: On page 5,6,7,15 and 16 you need to correct references to references, currently "Error! Reference source not found. "I suggest to improve (standardize) text formatting, eg page 11, 17. From page 15 the page numbering rule has been changed.
Response: Thank you very much for highlighting our mistake. We have corrected all the references. And we have also improved the formatting according to the journal rules.
Reviewer 2 Report
The manuscript presented a framework of IoT-based Industrial Data Management System (IDMS) that can manage industrial data, control smart manufacturing and support online monitoring. The authors also presented a case study that manages regular data and urgent events. The paper presents a good topic, however, a serious revision is required. You have to consider the following comments:
- Almost half of the references are old, it might be replaced by recent references.
- In sections 3 and 4, some references are not found and return an error (Error! Reference source not found).
- The flowchart in section (4.3.4. Recovery Management) is not clear. More justifications are needed in the text, what are the preparations? Please add a detailed analysis of all components.
- The flowchart is captioned as Figure 1, need to be modified to Figure 5 instead.
- In section (, 5.1. Case Study: Water Treatment Plant) three references are not found (return error messages).
- Figure 6 is not clear, Consider re-draw it.
- Quantify your conclusion.
- Font size and spaces are different in some sections.
- Consider some recent publications in the domain of IoT and Mangement such as the following:
* F. Anjomshoa, M. Aloqaily, B. Kantarci, M. Erol-Kantarci and S. Schuckers, "Social Behaviometrics for Personalized Devices in the Internet of Things Era," in IEEE Access, vol. 5, pp. 12199-12213, 2017.doi: 10.1109/ACCESS.2017.2719706.
** M. Aloqaily, I. Al Ridhawi, H. B. Salameh, Y. Jararweh, Data and service management in densely crowded environments: Challenges, opportunities, and recent developments, IEEE Communications Magazine.
*** S. Otoum, B. Kantarci and H. T. Mouftah, "On the Feasibility of Deep Learning in Sensor Network Intrusion Detection," in IEEE Networking Letters.
G. Bloom, B. Alsulami, E. Nwafor and I. C. Bertolotti, "Design patterns for the industrial Internet of Things," 2018 14th IEEE International Workshop on Factory Communication Systems (WFCS), Imperia, 2018, pp. 1-10.
**** Iqbal R.,Butt T.,Shafiq O.,Talib M.,Umer T.:“Context-AwareData-DrivenIntelligentFramework for Internet of Vehicles”, IEEE Access, 6(1), 58182-58194, 2018.
- Consider security perspectives, as well:
* M. Aloqaily, S. Otoum, I. Al Ridhawi and Y. Jararweh,”An intrusion detection system for connected vehicles in smart cities”, Ad Hoc Networks, 2019,ISSN 1570-8705.
** S. Otoum, B. Kantarci and H. T. Mouftah, "Detection of Known and Unknown Intrusive Sensor Behavior in Critical Applications," in IEEE Sensors Letters, vol. 1, no. 5, pp. 1-4, Oct. 2017, Art no. 7500804. doi: 10.1109/LSENS.2017.2752719
*** Ibrahim Ghafir, Jibran Saleem, Mohammad Hammoudeh, Hanan Faour, Vaclav Prenosil, Sardar Jaf, Sohail Jabbar, Thar Baker: Security threats to critical infrastructure: the human factor. The Journal of Supercomputing 74(10): 4986-5002 (2018)
- The manuscript needs more organization.
Author Response
Response to the Reviewer-2 Comments
Ref. No.: jsan-481009
Title: Framework of an IoT-based Industrial Data Management for Smart Manufacturing
Point 1: Almost half of the references are old, it might be replaced by recent references.
Response: Thanks for mentioning the important point. We have replaced some of the old references and new references have been added additionally.
Point 2: In sections 3 and 4, some references are not found and return an error (Error! Reference source not found).
Response: Thank you very much for highlighting our mistake. We have corrected all the references. All changes have been highlighted in yellow color accordingly.
Point 3: The flowchart in section (4.3.4. Recovery Management) is not clear. More justifications are needed in the text, what are the preparations? Please add a detailed analysis of all components.
Response: We have explained the complete flowchart. Preparation process indicates that data is being normally transmitted from data management component to distribute data storage component. All changes have been highlighted in yellow color on page 12.
Point 4: The flowchart is captioned as Figure 1, need to be modified to Figure 5 instead.
Response: Thanks for mentioning our mistake. It has been modified accordingly.
Point 5: In section (, 5.1. Case Study: Water Treatment Plant) three references are not found (return error messages).
Response: We have corrected all the references. The changes have been highlighted in yellow color accordingly.
Point 6: Figure 6 is not clear, Consider re-draw it.
Response: We have re-drawn Figure 6 to make it clearer. The changes have been highlighted in yellow color on page 15.
Point 7: Quantify your conclusion.
Response: We have improved the conclusion part to quantify it. The changes have been highlighted in yellow color.
Point 8: Font size and spaces are different in some sections.
Response: We have corrected this issue by making the same size and space of the whole manuscript. The changes have been highlighted in yellow color on page 3, 11, and 17.
Point 9: Consider some recent publications in the domain of IoT and Management.
Response: Thank you very much for good suggestion. We have added some of the recent relevant publication in Iot and management domain. All new added publication references have been highlighted in yellow color in reference section.
Point 10: Consider security perspectives
Response: We have added security parameters and references in the manuscript. The changes have been highlighted in yellow color on page 9 and 13.
Point 11: The manuscript needs more organization.
Response: Thank you very much for good suggestion. We have organized the manuscript according the comments. The changes have been highlighted in yellow color.
Round 2
Reviewer 2 Report
The authors have not taken the suggested comments seriously. Their review has been done quickly. More precisely:
The conclusion is too long and it didn't quantify the numeric improvements.
Most if not all of the figures are done in low quality. If I zoom the picture, the image becomes blurry.
The suggested relevant work has not been taken into consideration (Comment 9 and 10)!
The business approach in Fig. 2 is not discussed. This is important since it tackles the industrial point of view.
The work called " Data and service management in densely crowded environments: Challenges, opportunities, and recent developments" is perfectly fit your data management approach. Still not compared with!
Justify your answer to every comment logically.
Author Response
Response to the Reviewer Comments
Ref. No.: jsan-481009
Title: Framework of an IoT-based Industrial Data Management for Smart Manufacturing
Point 1: The conclusion is too long and it didn't quantify the numeric improvements.
Response: The conclusion has been further summerized and numeric improvements have been mentioned in it. All changes have been highlighted in yellow color on page 19.
Point 2: Most if not all of the figures are done in low quality. If I zoom the picture, the image becomes blurry.
Response: Thank you for pointing out important point. We have improved the quality of all images. Some of them have been redrawn and others have been improved using an official software.
Point 3: The suggested relevant work has not been taken into consideration (Comment 9 and 10)!
Response: Most of the mentioned references have been considered this time. All changes have been highlighted in yellow color on pages 2, 4, and 5.
Point 4: The business approach in Fig. 2 is not discussed. This is important since it tackles the industrial point of view.
Response: Thank you for pointing out important point. We have discussed it in new version of the manuscript. All changes have been highlighted in yellow color on pages 6 and 7.
Point 5: The work called " Data and service management in densely crowded environments: Challenges, opportunities, and recent developments" is perfectly fit your data management approach. Still not compared with!
Response: I am sorry as I tried to find this article by all available sources, but I couldn’t. If you provide me this article, I will surely be compared it.